



# Preference and Willingness-to-pay analysis for an eco-engineering technology for floating wind turbines

Antoine Dubois[1,2], Pierre-Alexandre Mahieu[3], Alison Bates[4], Jenifer Meredith[4], Franck Schoefs[2,5]

[1]Nantes Université, École Centrale Nantes, CNRS GeM, UMR 6183, Nantes, France
[2]Nantes Université, CNRS, IUML, FR CNRS 3473, F-44000, Nantes, France
[3]LEMNA, Nantes Université, Nantes, France
[4]Buck Lab, Department of Environmental Studies, Colby College, Waterville, Maine, USA
[5]Nantes Université, ISOMer, UR2160, 2 rue de la Houssinière, BP 92208, F-44000, Nantes, France

*Correspondence to*: Antoine Dubois (antoine3.dubois@gmail.com)

**Abstract.** As France accelerates its offshore wind energy ambitions to meet decarbonization targets, floating offshore wind turbines (FOWTs) have emerged as a key technology. However, concerns about their ecological and social impacts remain prominent among coastal populations. This study investigates public preferences and

willingness-to-pay (WTP) for an innovative eco-engineering solution to be integrated into future floating wind farms: a multifunctional structure aiming at enhancing marine biodiversity, supporting artisanal fisheries and minimizing seabed disturbance. A discrete choice experiment (DCE) was conducted on 306 French residents across five coastal departments to quantify trade-offs and explore territorial variation in acceptability.

The DCE included four attributes: structure material (recycled or new steel), biodiversity gain, impact on local

fisheries revenue, and additional cost to electricity bills. Results from a Conditional Logit Model and WTP estimation reveal a generally high level of support for eco-engineering features with biodiversity and fishery co-benefits strongly valued. Only the "recycled steel" attribute showed significant territorial variation, with Bouches-du-Rhône respondents exhibiting a higher WTP for this attribute.

The study also showed that negative attitudes toward offshore wind power were significantly associated with a

higher likelihood of selecting the *status quo* scenario, even when ecological enhancements were present. The study underscores the importance of integrating social preferences into the early design of FOWT projects and demonstrates that eco-engineering can be a viable lever for environmental and social integration of these projects.

## 1 Introduction.

Over the past decade, the French government has adopted an ambitious trajectory to reduce its greenhouse gas

emissions and transition towards a low-carbon economy. Aligned with the European Green Deal and its own Climate and Energy Framework, the country has committed to achieving net-zero carbon emissions by 2050 (ADEME, 2024). To do so, France aims at producing 40% of its electricity from renewable sources by 2030, with offshore wind power emerging as a cornerstone of its future energy mix (Ministère de la Transition Écologique, 2024). In 2023, the French government updated its offshore wind deployment target to 45 GW by 2050: still, this

aim is an unprecedented leap given that only 1.5 GW had been commissioned by mid-2025. This implies that 43.5 GW (approximately 96.7%) remains to be installed over the next 25 years (Figure 1 & Table A1).

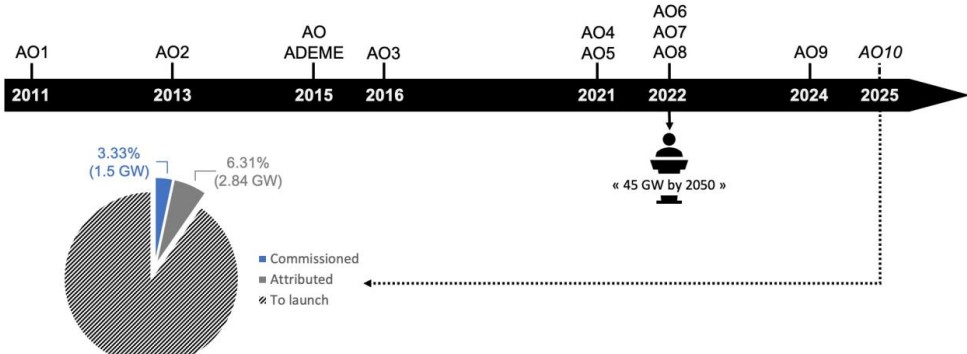

**Figure 1. Chronological order of call for tenders for Offshore Wind projects launched in France, and proportion of national goals achieved and remaining.**

Floating Offshore Wind Turbines (FOWTs) are emerging as a key technological and promising solution to meet the ambitions of France on offshore wind energy, particularly in regions where deep waters (> 60 m depth) preclude the use of bottom-fixed turbines. FOWTs allow wind farms to be located further offshore, reducing visual impact and expanding the potential surface area for renewable energy production (Zountouridou et al., 2015). However, they also present specific environmental and social challenges as well as an adaptation of harbors

infrastructures (Crowle and Thies, 2022). The increased depth and remoteness of FOWTs does not necessarily diminish public concerns. On the contrary, they may exacerbate anxieties about unknown ecological impacts, potential conflicts with fishing activities, and cumulative pressures on marine ecosystems (Chaumette, 2017; Dubois et al., 2025a; Jiang, 2021).

This technological shift and acceleration are reflected through the growing scale and ambition of national offshore

wind tenders. By the end of 2024, over 10 GW of capacity had entered formal procedures (Figure 1) and the upcoming "AO10" call for tenders will bring the total surface of wind exploitation in French metropolitan waters over 3,000 km². Among these "AO10"-projects are several large-scale floating wind farms each involving between 1 and 2 GW capacities (Table A1). These projects mark a clear shift towards industrial-scale deployment, both in terms of capacity and spatial footprint. This dynamic is coupled with a clear intention to maximize energy

production by leveraging the latest technological advancements. While the bottom-fixed Saint-Nazaire wind farm (awarded in 2012) used 6 MW turbines, more recent projects such as "*Bretagne Sud 1*", "*Golfe de Fos 1*" or "*Narbonnaise Sud-Hérault 1*" plan to deploy FOWTs of more than 20 MW per unit. However, implementation delays remain substantial (Table A1): on average, ten years elapse between the award of a project and the start of construction. For instance, construction of the wind farms projects awarded in the tenders "AO4", "AO5" and

"AO6", awarded respectively in 2023, 2024 and 2024 (Table A1), are not expected to begin until 2031–2032, but the concept of the turbine used is fixed when the tendered company is selected (*i.e.* around 5 years before the park is commissioned).

In this context, spatial planning and public acceptability have become central challenges for the successful development of offshore wind energy (Joalland and Mahieu, 2023). Numerous studies have demonstrated that

public opposition to offshore wind farms can be driven by a variety of factors such as visual and landscape concerns (Ladenburg, 2010), place attachment (Brownlee et al., 2015), perceived procedural fairness and justice (Bacchiocchi et al., 2022; Firestone et al., 2012) or even trust in institutions (Druckman, 2015; Handmaker et al.,





2021). Impacts on marine biodiversity is also frequently cited as a major concern (Bush and Hoagland, 2016; Galparsoro et al., 2022). As a result, developers are increasingly required to implement early-stage environmental

monitoring, including the assessment of acoustic pollution, benthic disturbances and interactions with marine mammals (Degraer et al., 2021; Maxwell et al., 2022). Furthermore, these studies and monitoring are part of the process of verifying the proper integrity of structures throughout the farm's service life (Coolen et al., 2018; Coughlan et al., 2025; Dubois et al., 2025b).

Eco-engineering is increasingly being explored as a potential lever for reconciling technological development

with ecological integrity. This concept refers to the design and implementation of infrastructure in a way that integrates ecological functions and enhances ecosystem services (Pardo et al., 2023). In the context of offshore wind, this may involve integrating habitat-enhancing structures directly into wind farm components such as moorings, scour protection or substations to promote biodiversity and ecosystem functioning (O'Shaughnessy et al., 2020). Recent frameworks such as Nature-Inclusive Design (NID) and marine Nature-Based Solutions (NbS)

have strengthened the case to integrate such approaches directly within development process, particularly to respect the "avoid–reduce–compensate" hierarchy used in marine spatial planning (Hermans et al., 2020; Sutton-Grier et al., 2015). These concepts aim not only to reduce ecological harm but also to generate measurable co-benefits for marine ecosystems and local stakeholders. In practice, eco-engineering is the modification of structures with the use of artificial reefs, textured concrete modules or biologically active substrates, or also the

inherent design of structures to attract reef-associated species, stabilize sediments or create nursery habitats (Firth et al., 2014; Lengkeek et al., 2017). However, as Bishop et al. (2017) pointed it out, the ecological success of such measures depends heavily on spatial scale, species-specific requirements and physical compatibility.

Eco-engineering is being recognized as a social as well as technical innovation, raising important questions about governance, legitimacy and the role of local communities in defining what constitutes acceptable and meaningful

ecological compensation (Dennis et al., 2018; O'Shaughnessy et al., 2020; Varenne et al., 2023), especially in the context of ocean sprawl or Non-Indigenous Species facilitation (Gauff et al., 2023).

Research suggests that such measures may improve social acceptance, particularly when they generate perceived co-benefits (Klain et al., 2020; Strain et al., 2019). Moreover, recent qualitative research by Dubois et al. (2025a), comparing coastal community perceptions in France (Pays de la Loire) and the United States (Maine), highlights

the complexity of public attitudes towards the new technological concepts of floating offshore wind farms combined with artificial reef structures. While participants generally supported the energy transition, persistent concerns were expressed regarding environmental impacts (biodiversity, seascapes), economic consequences (fisheries, tourism) and technical uncertainties (cost, maintenance). These findings underline the importance of transparent, participatory and science-based governance in reconciling climate goals with the social expectations

of coastal communities characterized by diverse identities and competing uses.

The present study investigates public preferences for an innovative eco-engineering solution specifically designed for integration into floating offshore wind farms. The solution takes the form of a multifunctional artificial structure intended to simultaneously (i) increase local biodiversity, (ii) support artisanal fishing and (iii) reduce the ecological footprint of FOWTs by limiting seabed dragging caused by mooring lines. As a hybrid between

ecological compensation and technical optimization, this innovation tries to embody a model of spatial and functional cohabitation that could help mitigate stakeholder opposition and contribute to the long-term viability of floating wind deployment.



While existing studies have explored how environmental attributes influence public preferences for wind energy, few have examined the acceptability of integrated technological and ecological innovations into such technology. Moreover, no previous study has assessed such preferences in the specific context of France's floating offshore wind strategy. This study addresses that gap by implementing a Discrete Choice Experiment (DCE) targeting a representative sample of 306 coastal residents across five French departments with various cultural, economic and industrial relationship to the sea. The DCE includes four key attributes: (i) the material used for building the structures, (ii) expected augmentation in marine biodiversity (specific richness), (iii) anticipated economic effects on the local-artisanal fisheries and (iv) a cost attribute to estimate willingness to pay (WTP). This approach allows to quantify trade-offs in citizen preferences and explore variation in acceptability across regions and individual profiles.

More specifically, the primary objective of this study is to identify the preferences of citizens from five French departments regarding an integrated offshore eco-engineering solution, and to test whether social acceptability varies across territories and individual attitudes. Thus, this territorial comparison is designed to test whether public preferences vary across coastal contexts. While the null hypothesis assumes no significant differences between departments, we expect that local factors (such as dependence on fisheries or exposure to existing offshore projects) may influence the acceptability of eco-engineering solutions. Identifying these variations can support more tailored and socially informed planning strategies. Another possible hypothesis to make is that offshore wind opponents will try to minimize environmental or social impacts by selecting projects that included mitigation measures.

The paper is organized as follow: section 2 describes the tested concept of eco-engineering, the method used to analyze its societal acceptability and the territorial identities of the 5 selected departments. Section 3 presents the results of the willingness-to-pay for the application of the concept and the parameters influencing on the choice of scenario. Section 4 discusses about the results depending on the department studied and the effect of attitude toward offshore wind power on the application of an eco-engineering concept. Section 5 is dedicated to developers and industry stakeholders for future offshore wind power development and section summarize the findings of the study.

## 2 Material & Method.

### 2.1 The eco-engineering concept.

In our study we focus on a concept designed specifically for application in floating offshore wind farms. After discussions in a previous study (Dubois et al., 2025a), we targeted the respondents' priorities and concerns to help in the design of this structure. In the end, the concept was a stack of steel pipes of various diameters. Despite the paucity of information on this subject, some sources indicate an optimal volume for an artificial reef in OWF of the order of 320 m³ (Glarou et al., 2020; Langhamer, 2012). Thus, the theoretical volume of this concept is 400 m³ with a steel volume of 43.5 m³. Together with the increase in biodiversity and biomass, the structure fits into the framework of eco-engineering (Hermans et al., 2020; Pardo et al., 2023; Pioch et al., 2018) by limiting the seabed dragging by the mooring lines. This could be achieved by passing the mooring line through the center of the structure, thus shifting the line upwards so that it does not touch the seabed. For the chain, this reduces wear and tear, and for the environment, this considerably reduces chain slippage on the floor as the float moves. This type of structure would then be used above each floating wind turbine anchor on a wind farm. Overall,





they would serve a triple purpose: 1) they would limit the footprint of a farm, 2) they would provide an opportunity for refuge and habitat creation, and 3) they would have an impact on society in terms of both societal acceptability and the economy (*e.g.* fishermen).

**2.2 Survey design.**

**2.2.1 DCE method.**

This study uses the Discrete Choice Experiment (DCE) method to identify individuals' preferences (Hoyos, 2010) and estimate their willingness-to-pay (WTP) for different characteristics of a good or service (Hanley et al., 1998). Based on the theory of random utility (McFadden, 1974), this approach relies on the analysis of choices made between several alternatives defined by combinations of attributes. DCE was chosen for its ability to quantify trade-offs between attributes and to incorporate a payment vehicle enabling direct monetary estimation. Its implementation in digital format also facilitates large-scale dissemination, enables a large and geographically diverse sample. This method has several advantages: theoretical soundness, applicability to non-market goods, and the ability to model preference heterogeneity. However, it has certain limitations, including a potential cognitive burden for respondents, sometimes questionable rationality assumptions and sensitivity to formulation or fatigue biases. Despite these constraints, DCE remains a benchmark method for preference analysis and the economic evaluation of goods and services.

**2.2.2 Geographical Sampling.**

An online national survey was performed through a market company in April 2024. Five French departments were aimed, depending on their proximity to planned development of Floating Offshore Wind Farm. These departments were the following: Aude, Bouches-du-Rhône, Hérault, Morbihan, Pyrénées-Orientales. Four of these departments board the coast of the French Mediterranean Sea and one (Morbihan) is at the coast of French Brittany and boarded by the Atlantic Ocean. The total number of respondents was 306 and sampling was carried out in such a way as to obtain proportions as representative as possible to the number of inhabitants in each department, with 20 respondents from Aude, 114 from Bouches-du-Rhône, 87 from Morbihan, 54 from Hérault and 31 from Pyrénées-Orientales.

We drew up the territorial identities of the sample departments (Table 1), in relation to the subject of study, taking into account: demographic and blue economic statistics including tourism (touristic rate being the number of touristic beds on the number of residents in the department), fishing and industry, information on ecology (through protected areas) and fishing (share of maritime employment and tons of seafood landed). The percentages are expressed in function of the department level.

**2.2.2.1 Aude: a discreet coastline between tourist appeal and economic fragility.**

Literature and data obtained for the Aude department generally point to limited maritime employment in this territory, with 3.1% of employees working in the maritime sector (INSEE, 2017) and 1,600 tons of seafood products landed (FranceAgriMer, 2024) for a population of 376,000 persons (INSEE, 2025a). However, commercial tourism is highly structuring the economy of the department, with a high rate of second homes (25.3% of departmental homes) and a relatively high tourist function rate with 42.82% (Agence de Développement





Touristique, 2024). Finally, the maritime domain is still little preserved with around 6% of its surface area under any protection regime (Les sites Natura 2000 dans l'Aude, 2019).

**2.2.2.2 Bouches-du-Rhône: A strategic, industrial-port coastline and less tourist.**

The Bouches-du-Rhône department is one of the most urbanized and densely populated French coastal areas with more than 2 million residents (INSEE, 2025b), but with limited tourism according to the touristic rate of 18% and 4.8% of secondary homes (Observatoire en ligne Provence Tourisme, 2025). The area is also characterized by a strong maritime presence, but rather focused on logistics and industry than fishing with around 3,833 tons landed

per year (Ifremer, 2024b). It hosts the second most important commercial harbor in France (Marseille-FOS). On the other hand, there are some real natural gems, such as the "Côte Bleue" Marine Park and the Calanques National Park, which are considered as true marine sanctuaries thanks to zones reserved from human impact (diving, fishing), bringing the surface area of protected marine areas (all statuses combined) to around 45,000 ha (Bottin, Garcia and Meinesz, 2020) and almost 10,000 ha fully preserved of anthropic activities.

**2.2.2.3 Hérault: A dense, multifunctional coastline, between tourism and the blue economy.**

Hérault department is characterized by a dense population of more than 1.2 million residents (INSEE, 2025c), and a single harbor structuring the marine employment that is located at Sète where the fishing landings are concentrated and cumulating around 7,146 tons of seafood annually (Ifremer, 2024c). The proportion of maritime employment in the departmental activity is around 4.4%. This department is highly attractive for tourism,

especially the seaside tourism, and this activity represents a great part of the local economy highlighted by the tourist function rate of 83% and the proportion of secondary residences of 17.8% (INSEE Flash Occitanie, 2022; Chiffres clés Tourisme et Loisirs Hérault édition 2024, 2025). Also, with 8,500 ha of marine protected areas, this department is in the process of reconciling tourism with the protection of its natural heritage (Bottin, Garcia and Meinesz, 2020).

**2.2.2.4 Morbihan: a coastline balanced between maritime traditions and tourist appeal.**

The Morbihan department is bordered by the Atlantic Ocean and therefore has a very different history from the Mediterranean departments. This Breton department has one of the highest maritime employment rates in France with more than 7% (Février and Le Guen, 2018), for a population above 760,000 residents (INSEE, 2025d). At the same time, the fishing industry in Morbihan is one of the main sectors in the local economy with almost 22,000

tonnes of seafood products landed each year (Ifremer, 2024d). On top of this, the area is a major draw for tourists thanks to its culture and landscapes with a high tourist function rate of 85% and 17.8% of secondary residences. Another attraction for tourism is the balance between maritime exploitation and preservation in the Gulf of Morbihan, with some 70,000 ha of protected marine areas (DREAL Bretagne, 2023).

**2.2.2.5 Pyrénées-Orientales: A hyper-touristic coastline with a modest maritime profile.**

Last but not least, the Pyrénées-Orientales department lies midway between mountain ranges and coastlines, making it an attractive location for tourism. The population is modest with almost 490,000 residents. This is reflected in the tourism offer, particularly in the tourist function rate of 132% (Capacité d'accueil Pyrénées Orientales Tourisme, 2025) and a high rate of 27.7% of second homes (INSEE, 2025e). Tourism is thus the



mainstay of the local economy. On the other hand, maritime activity is more limited, with a low proportion of
maritime employment (3.7%, INSEE, 2017) and more limited landings than other departments (1,501 tons/year
Ifremer, 2024e). The documentation found estimates at around 11,000 ha the surface of marine protected areas of
any status (Bottin, Garcia & Meinesz, 2020; De Paoli et al., 2023).

**Table 1. Identities of the sampled territories (departments) subject to floating offshore wind development.**

| Metrics | Aude | Bouches-du-Rhône | Hérault | Morbihan | Pyrénées-Orientales |
|---|---|---|---|---|---|
| Number of residents | 376,028 | 2,056,943 | 1,201,883 | 768,687 | 487,307 |
| Proportion of maritime employment | 3.1% | 4.4% | 3.2% | 7.4% | 3.7% |
| Tons of seafood landed per year | 1,624 tons | 3,833 tons | 7,146 tons | 22,607 tons | 1,501 tons |
| Tourist function rate (nb of tourist beds/residents) | 42.82% | 18% | 83% | 85% | 132.52% |
| Proportion of secondary residences (departmental) | 25.3% | 4.8% | 17.8% | 17.8% | 27.7% |
| Surface of marine protected areas (any status) in acres | 34.5 acre | 111k-123k acre | 21k acre | 173k acre | 27k acre |
| Marine High-protection zone in acres | 0 acre | 23k acre | 0 acre | 32.5 acre | 247 acre |


### 2.2.3 Organisation of the survey.

The questionnaire started with socio-demographic questions: place of residency (zip code), education level, actual
employment status and revenue after taxes and per month (France). These questions were followed by the choice
experiment. Before the series of choices, an introduction was included with the following information:

1.     Electricity mix in France and governmental goals.

     2.     Explanation of a FOWT and what is the situation in their country.

     3.     Explanation of the reasons of going towards a FOWT development.

     4.     Goals about this technology development, comparison with nuclear power and number of household's
electricity consumption.

5.     Impacts of FOWT (environmental, economic).





6.      Presentation of the eco-engineering concept with visualizations.

7.      Explanation of how a DCE works and description of each attribute with their meanings.

8.      Explanation of the status quo.

After the choice experiment, respondents were asked several follow-up questions about their perception of offshore wind power attitudes, their relation with the ocean (any relatives working with/depending on the ocean and/or fishing), having heard or seen an OWF before this survey and finally a New-Ecological Paradigm test was performed through a likert-scale questionnaire with 15 questions (Appendix 2; Anderson, 2012; Dunlap et al., 2000). These parameters were implemented into a correlation test after the econometric model. The zip code of residency allowed to calculate the average distance from the coast, the department of the city and the region of

the city.

### 2.2.4 The *status quo* scenario.

The *status quo* scenario chosen reflects France's current trajectory in offshore wind power: rapid, intensive development of wind farms, with no particular requirements beyond the regulatory framework imposed. It corresponds to floating wind farm projects that could be described as "classic", with no specific eco-engineering

measures, apart from the environmental monitoring required before and after commissioning and throughout the service life of the farm until decommissioning. This scenario serves as a realistic reference point, consistent with national guidelines, and enables to measure preferences for alternatives that incorporate greater ecological ambitions.

### 2.2.5 The attributes and their levels.

The attributes were chosen on the basis of a preliminary study in which respondents expressed their fears and priorities with regard to the development of offshore wind power, whether bottom-fixed or floating (Dubois et al., 2025a). Moreover, literature was taken into consideration to scale the levels of chosen attributes (Börger et al., 2015; Dalton et al., 2020; Iwata et al., 2023; Kermagoret et al., 2016; Kim et al., 2019; Klain et al., 2020). The definition of levels for each attribute is based on a combination of findings from the scientific literature, empirical

data from fisheries and energy reports, and adjustments based on pre-testing of the questionnaire. The aim was to propose realistic, credible and comprehensible levels for respondents, while ensuring sufficient variability to capture differentiated preferences.

#### 2.2.5.1 Structure material.

The material used for the structure (recycled or new steel) is a central environmental indicator. Recycled steel has

a 20-25% lower carbon footprint than new steel (Fennell et al., 2022), with an emissions reduction potential of 1.5 ton of $CO_2$ per ton of steel (World Steel Association, 2021). France already produces around 40% of its steel from recycled materials (CNDP, 2024), making this attribute both credible, measurable, and culturally relevant. It also makes it possible to test citizens' sensitivity to aspects of circularity in energy infrastructures.

#### 2.2.5.2 Impact on marine biodiversity.

The biodiversity attribute was defined on the basis of extensive literature on the effects both of offshore wind farms and artificial reefs. Several studies demonstrate a local increase in biodiversity due to the reef effect





phenomenon, where submerged structures (foundations, cables, floats) promote colonization by fixed species such as mussels, anemones, algae or soft corals (Andersson and Öhman, 2010; Coolen et al., 2018; Degraer et al., 2021; Dubois et al., 2025a). Rates of increase in biodiversity ranging from 10% to 200% have been reported depending

on the context (Brock and Norris, 1989; Fabi and Fiorentini, 1994), although the range generally adopted in previous DCE varies between 10% and 60% (*e.g.* Klain et al., 2020). In order to remain within a zone of ecological plausibility and facilitate understanding for respondents, the following four levels were retained: +10%, +20%, +30% and +40% increase in marine biodiversity on average throughout the service life of the farm. This increase refers to the increase in species richness "S" (Anon, 2009). The experimental design was inspired by previous

work carried out on artificial reefs where the addition of hard substrates has demonstrated strong potential for biological colonization (Koeck et al., 2014; Komyakova et al., 2021). The structures studied were modeled with a volume of around 320 m³ (Glarou et al., 2020), the optimum size suggested in the literature to maximize ecological effects.

### 2.2.5.3 Impact on local fisheries revenue.

The effect of floating wind turbines on fishing was addressed through changes in local fishermen's income, an indirect but relevant indicator for respondents (Bates and Firestone, 2015; Firestone and Kempton, 2007). Based on studies of fishing yields around artificial reefs (CPUE - Catch Per Unit Effort), a link was established between an increase in biomass and biodiversity, combined with a potential increase in catches. A literature review (De Backer and Hostens, 2019; Ramos et al., 2006; Reubens et al., 2013) was used to translate CPUE gains into

economic impacts. A 60% catch-to-revenue conversion was adopted on the basis of existing data (Pan, 2021), then reduced to take account of operational constraints (closed areas, affected ports, etc.). The estimated impact was refined by cross-referencing windfarm development zones with data from the main fishing ports in the French Gulf of Lion (Ifremer, 2024a). To include differentiated but plausible scenarios, and following the pre-test highlighting the absence of an "extreme" case, the levels retained were: +1%, +5%, +10% and +15% increase in

fishing income in the zones concerned and on average throughout the service life of the wind farm.

### 2.2.5.4 Cost to households - electricity bill.

The last attribute is monetary and represents the monthly extra cost on the electricity bill induced by the integration of eco-engineering structures in wind farms. This cost was estimated by modeling the price of steel structures (320 m³ total volume, 43.5 m³ steel) from computer aided designing and its installation offshore, then integrated

into electricity production costs via an economic simulator (Energy101, 2025). Standard parameters were considered for floating wind farms, including a capacity of 1,050 MW, a capacity factor of 60%, a lifespan of 20 years and an interest rate of 6%. Three consumption profiles were simulated (1 person, 2 people, 4 people respectively in a studio, a small apartment or in a house), with amounts ranging from +€0.40 to +€7.76 per month depending on the profile. In addition to these estimates, feedback from the pre-test suggested the inclusion of a

higher cost level to capture economic trade-offs. Thus, five levels have been retained: +1 €, +2 €, +3 €, +5 € and +10 € per month, over a 20-year period and for a household. This attribute also plays the role of payment vehicle in the willingness-to-pay (WTP) estimation. These values were in line with previous research (Kim et al., 2019; Krueger et al., 2011).



**2.3 Econometric models.**

**2.3.1 Conditional Logit Model & Willingness-To-Pay (WTP) estimation.**

The analysis conducted in this study relies on the Random Utility Theory maximization approach (McFadden, 1974). When a respondent chooses a scenario for a FOWF development, the respondent is supposed to choose the option that maximizes the satisfaction that is derived from the attributes and their levels. The utility function is as follows Eq (1):

$U_{nj} = \beta x_{nj} + e_{nj}.$            (1)

For each respondent $n$, any alternative of floating wind farm development $j$ is associated with a specific level of utility $U_{nj}$, where the utility level depends on the vector of attributes of the scenario $x_{nj}$, which are here in this study: the material of the reef, the variation of biodiversity, local revenue as well as the monthly electricity bill. The term $\beta$ is a vector of preference parameters associated with the observed attributes of an alternative. The error

component $e_{nj}$ is composed of the unobserved characteristics that influence the decision-making of individual $n$ meaning that predictions cannot be output with certainty.

The marginal WTP, also called the implicit price, can be estimated for each of the non-cost attribute as follows, as explained by Hanley et al. (1998), where $\beta_c$ is the coefficient of any of the attribute and $\beta_y$ is the coefficient of the cost attribute (which corresponds to the marginal utility of income), Eq (2):

$WTP = -\beta_c / \beta_y$            (2)

After estimating the Conditional Logit Model, a Wald test was performed to evaluate the joint significance of the selected explanatory variables (Greene, 2019; Woolridge, 2010). The Wald test is computed from the estimated coefficients and their covariance matrix and follows an asymptomatic chi-square distribution under the null hypothesis that the tested parameters are equal to zero. It allows to test whether groups of variables contribute

significantly to the explanatory power of the model.

**2.3.2 Zero-Inflated Negative Binomial regression model to explain the choices**

A zero-inflated negative binomial (ZINB) regression model was used to analyze the determinants of respondents' tendency to choose the *status quo* option. The dependent variable ('*Number of status quo chosen*') represents the number of times each respondent selected the *status quo* across the eight choice scenarios performed by the

respondent. Preliminary inspection of the distribution revealed a large proportion of zeros indicating that many respondents never chose the *status quo* option. This overdispersion and excess of zeros (Cameron & Trivedi, 2013) makes traditional ordinary least squares (OLS) regression unsuitable, as it assumes normally distributed residuals and constant variance. Preliminary OLS models confirmed the lack of fit and heteroscedasticity.

The ZINB model decomposes the data-generating process into two parts (Hilbe, 2011, 2014; Yau et al., 2003): (i)

a count model, which predicts the number of status quo choices for respondents capable of choosing it, modelled using a negative binomial distribution, and (ii) a zero-inflation model, which predicts the probability that a respondent always chooses zero (*i.e.* never selects the *status quo*), modelled with a logistic regression. The count part included the following covariates: attitude toward floating offshore wind power, stated gender, age, level of education, professional status, monthly revenue, prior exposure to offshore wind power projects (have already

seen or heard about offshore wind turbines), environmental attitudes (through the NEP mean score), relationship



to the ocean (having a relative working with the ocean), fishing activity (having a relative that is a commercial fisher) and finally the distance to the coast in kilometres.

Model selection was informed by comparisons of Akaike Information Criterion (AIC) and Bayesian Information Criterion (BIC) across alternative specifications, including Poisson, negative binomial, zero-inflated Poisson (ZIP) and ZINB models. The ZINB model was selected as the most appropriate due to its superior fit (lowest AIC and BIC in Table 2) and ability to accommodate both overdispersion and excess zeros (Greene, 1994; Hall, 2000).

**Table 2. Akaike Information Criterion and Bayesian Information Criterion for optimal selection of model.**

| Model | AIC | BIC |
|---|---|---|
| Poisson | 1380.654 | 1429.061 |
| Zero-inflated Poisson | 931.968 | 987.821 |
| Zero-Inflated Negative Binomial | 913.716 | 973.293 |
| Negative Binomial | 945.833 | 997.963 |

All analyses were conducted in R (version 4.3) using the "pscl" package (Jackman, 2024) for zero-inflated models. Standard errors and statistical significance were derived from the model summary output and incidence rate ratios (IRRs) were calculated by exponentiating the coefficients from the count model to aid interpretation.

## 3 Results of Willingness-to-pay for an eco-engineering concept.

### 3.1 Descriptive statistics.

The sample is characterized by a departmental profile contrast in comparison with the national average (Table 3). Bouches-du-Rhône sample stands out with younger respondents, a high activity rate (75%), a high proportion of high education (41% at least bachelor) and an average net income well above the national average (€3,100 vs. €2,336). Conversely, Aude sample has an older population, lower levels of education (30%), lower activity rate (50%) and lowest average income (€2,000). Morbihan and Hérault samples present intermediate profiles with average incomes but an older population (especially in Morbihan) and relatively low graduation rates. Lastly, Pyrénées-Orientales has a high income but a more masculine structure and moderate activity levels.



**Table 3. Socio-demographics data from the samples; *data from 2019; ** data from 2017 (INSEE, 2020)**

| Variables | Aude | Bouches-du-Rhône | Hérault | Morbihan | Pyrénées-Orientales | France |
|---|---|---|---|---|---|---|
| Mean age | 54.5 (+/- 14.12) | 50.57 (+/- 12.38) | 51.59 (+/- 13.81) | 54.68 (+/- 12.20) | 53.35 (+/- 14.97) | *NA* |
| Proportion female | 0.45 | 0.5175 | 0.5517 | 0.5741 | 0.4194 | 0.517* |
| Income (monthly, net, after taxes) | 2000 € (+/- 877.35) | 3100.88 € (+/- 1498.79) | 2689.66 € (+/- 1318.45) | 2875 € (+/- 1188) | 2927.42 € (+/- 1453.72) | 2335.83 €** (+/- 3791.66) |
| Education (at least bachelor or equivalent) | 30.00% | 41.23% | 27.59% | 22.22% | 35.48% | 23.6%* |
| In professional activity (employed or independent) | 50% | 75.44% | 55.17% | 57.41% | 54.84% | 65.5%* |
| Observations | 20 | 114 | 87 | 54 | 31 | *NA* |


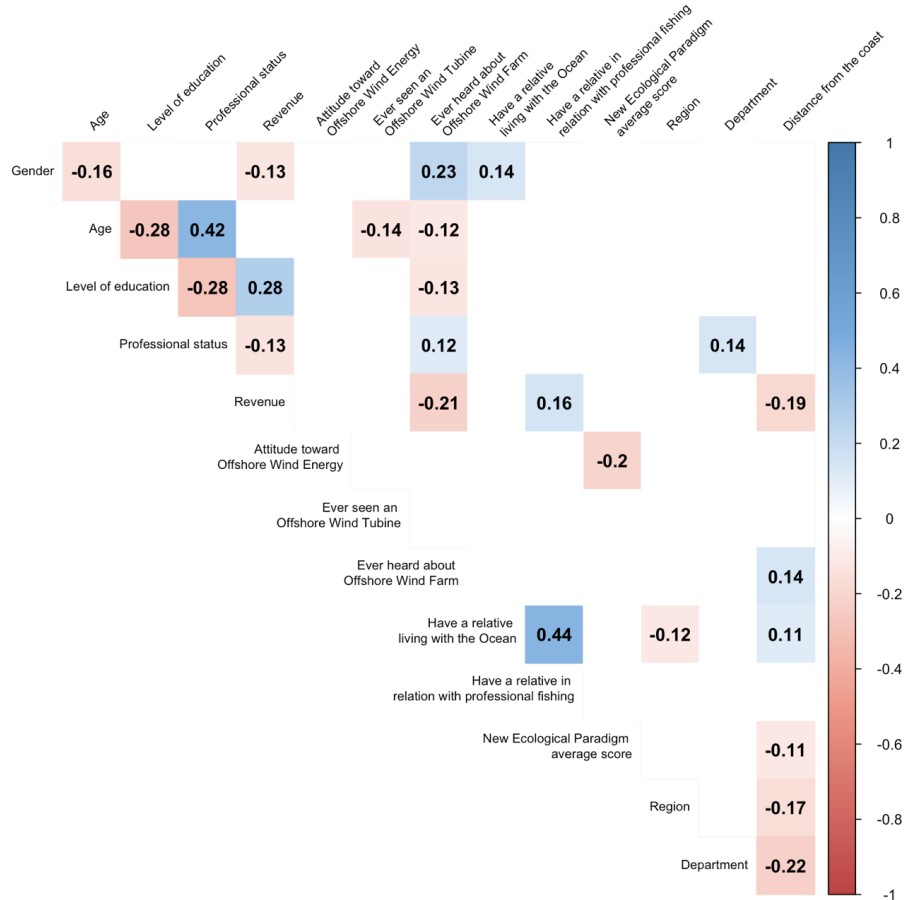

**Figure 2. Correlation test on socio-economical parameters of the samples.**

The coding of response allowed the run of a spearman correlation test on the socio-economical parameters of the

participants. Figure 2 presents the significant correlations (p-value < 0.05). Even though relatively weak, several

notable positive significant correlations were found, including:

- o   professional status and having heard about OWF (0.12).
- o   declared gender of the respondent and having heard about OWF (0.23).
- o   declared gender and having a relative working with the ocean (0.14).

- o   having heard about OWF with distance to the coast from the city of residency (0.14).

At the same time, several notable negative significant correlations were found such as:

- o   declared gender with the monthly revenue (-0.13).
- o   level of education and having heard about OWF (-0.13).
- o   the monthly revenue and having heard about OWF (-0.21).

- o   the stated attitude toward OWF and the average score of the New Ecological Paradigm (-0.2).





o the average score of the New Ecological Paradigm with distance to the coast from the city of residency (-0.11).

**3.2 Conditional Logit Model: relative importance relative of attributes per department.**

**Table 4. Coefficients (and robust standards errors) from the Conditional Logit Model.**

| Attributes | Aude | Bouches-du-Rhône | Hérault | Morbihan | Pyrénées-Orientales |
|---|---|---|---|---|---|
| Recycled steel | **0.652** (0.271)** | **0.589*** (0.129)** | 0.152 (0.138) | **0.293* (0.177)** | 0.262 (0.256) |
| Increase of Specific richness | **0.028** (0.009)** | **0.026*** (0.004)** | **0.024*** (0.004)** | **0.013** (0.006)** | **0.026*** (0.007)** |
| Local fishing revenue growth | **0.031* (0.018)** | **0.033*** (0.008)** | **0.037*** (0.01)** | **0.046*** (0.012)** | **0.035** (0.014)** |
| Increase of renewable-based electricity bill per month | **-0.165** (0.057)** | **-0.235*** (0.027)** | **-0.225*** (0.033)** | **-0.203*** (0.038)** | **-0.216*** (0.057)** |
| Notes | *p-value<0.05; **p-value<0.01; ***p-value<0.001 | | | | |

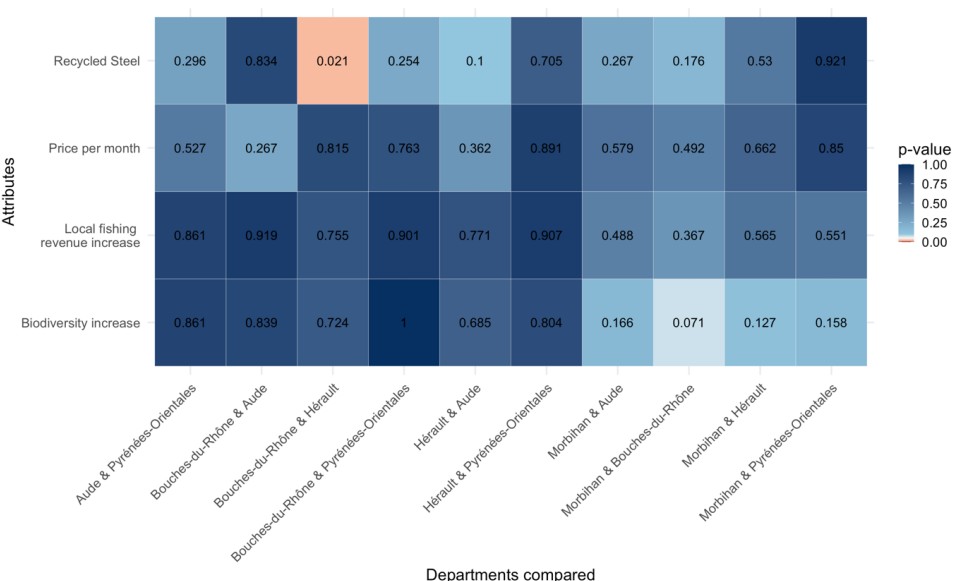

**Figure 3. Conditional Logit Model coefficient comparison (Wald Test) between department in function of attribute.**





The conditional logit model carried out on the data according to department shows significance for practically all the factors taken into account (Table 4). Only the 'Recycled steel' factor is not significant for the departments of Hérault and Pyrénées-Orientales. The results thus indicate the sensitivity of respondents to the attributes and their levels. The payment attribute (electricity bill) is the only one to have negative coefficients, indicating a limitation

of the increase in values for this attribute by respondents.

A Wald test was performed to analyze the presence or absence of differences in attributes between sampled departments (Figure 3). Only the 'Recycled Steel' attribute between the Bouches-du-Rhône and Hérault departments was significantly different. Despite the absence of statistical evidence (p-value > 0.05), the attribute 'Increased biodiversity' between the Morbihan and Bouches-du-Rhône departments is notable.

**3.3 Estimated Willingness-To-Pay (WTP).**

**Table 5. Estimated Marginal WTP coefficients from the Conditional Logit Model.**

| Attributes | Bouches-du-Rhône | Hérault | Morbihan | Aude | Pyrénées-Orientales |
|---|---|---|---|---|---|
| Recycled steel | **2.51\* (0.62)** | 0.68 (0.59) | 1.44 (0.9) | **3.94\* (1.88)** | 1.21 (1.14) |
| Increase of Specific richness | **0.11\* (0.02)** | **0.11\* (0.02)** | **0.06\* (0.03)** | **0.17\* (0.08)** | **0.12\* (0.04)** |
| Local fishing revenue growth | **0.14\* (0.03)** | **0.16\* (0.04)** | **0.22\* (0.06)** | 0.19 (0.11) | **0.16\* (0.06)** |

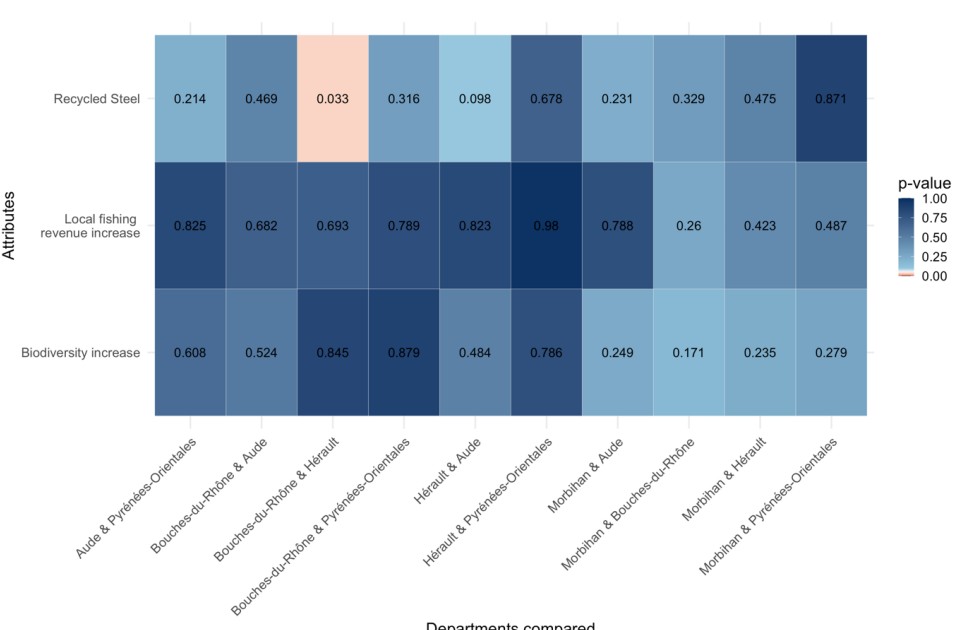

**Figure 4. WTP (Wald Test) between each department in function of the attributes.**



The estimation of WTP revealed a large majority of significant coefficients (Table 5). The coefficient for the attribute 'Recycled steel' is not significant for the departments of Hérault, Morbihan and Pyrénées-Orientales.
The same case is found for the attribute 'Growth in local fishing revenues' for the Aude department.

A Wald test was performed to analyze whether or not there was a significant difference between departments for an attribute (Figure 4). This test revealed a single significant difference between the coefficients derived from the Conditional Logit Model for Recycled Steel between respondents from Bouches-du-Rhône and Hérault (p-value < 0.05). Similarly, the marginal WTPs were analyzed with this Wald test, and the same result emerged: only the
marginal WTP for Recycled Steel was significantly different between respondents from Bouches-du-Rhône and Hérault.

**3.4 Attitude towards offshore wind power: a global point of view rather than territorial.**

A Chi² test was performed to assess whether the respondents' departments of origin had an effect on their attitudes towards offshore wind power (Figure 5). The results of this analysis showed no significant difference between
departments in attitudes (simulated Chi² B10000, p-value > 0.05). In an attempt to discern a trend, an identical test was carried out, grouping 'Very Positive' with 'Positive', and 'Very Negative' with 'Negative': the results of this test were also unsuccessful to detect differences (simulated Chi² B10000, p-value > 0.05).

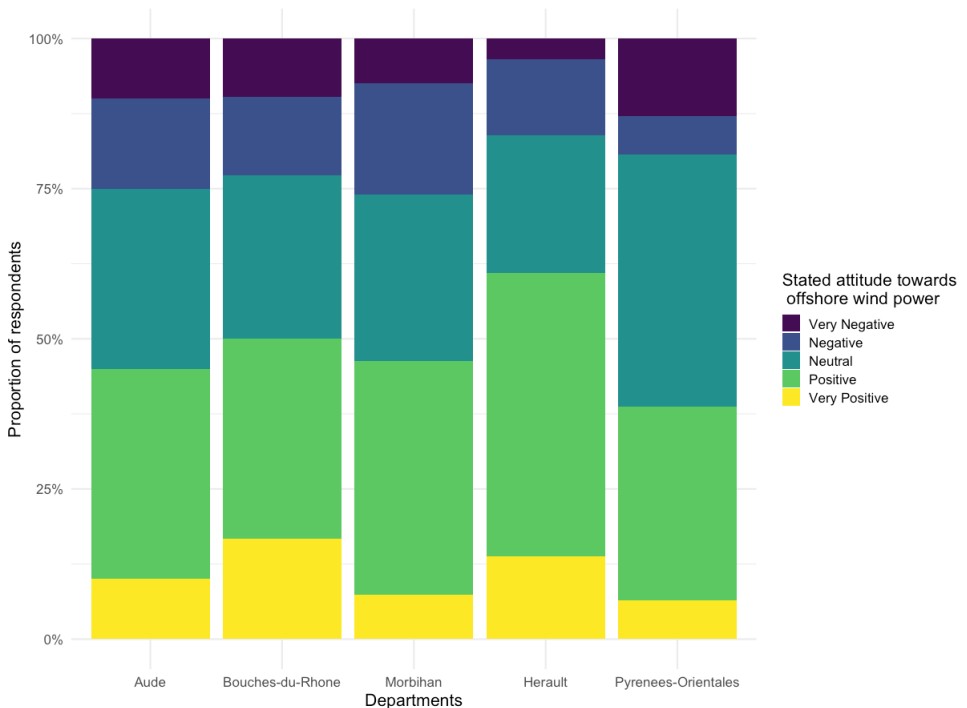

**Figure 5. Proportion of each attitude toward offshore wind power depending on the department.**





### 3.5 Link between stated attitude towards offshore wind power and frequency of chosen *status quo*: Zero-Inflated Negative Binomial regression model.

The frequency of *status quo* choices was dichotomized into two categories to facilitate interpretation and visualization: respondents selecting the *status quo* more than four times out of eight (> 50%) were classified as "Often", while those selecting it four times or fewer (equal or less than 50% of the time) were classified as "Rarely". This threshold was chosen to capture a meaningful distinction between systematic reliance on the *status quo* versus more occasional selection, while ensuring balanced group sizes for comparison. People with a 'Very negative' attitude often (15 out of 24; Table 6 and Figure 6) chose the *status quo*. Conversely, those with a more positive attitude ('Positive' or 'Very positive') rarely chose the *status quo* on a systematic basis (147 times vs. 9). It should be pointed out that the *status quo* did include the creation of a floating wind farm, but without the addition of the eco-engineering system. However, it is also notable that those with a declared 'Negative' attitude towards offshore wind power mostly rarely selected the *status quo* (30 times vs. 11).

**Table 6. Number of *status quo* chosen in function of stated attitudes by respondents.**

| Stated Attitudes | ≤ 4 times on 8 status quo chosen | > 4 times on 8 status quo chosen | **Total** |
|---|---|---|---|
| Very negative | 9 | 15 | **24** |
| Negative | 30 | 11 | **41** |
| Neutral | 71 | 14 | **85** |
| Positive | 114 | 3 | **117** |
| Very positive | 33 | 6 | **39** |
| **Total** | **257** | **449** | **306** |

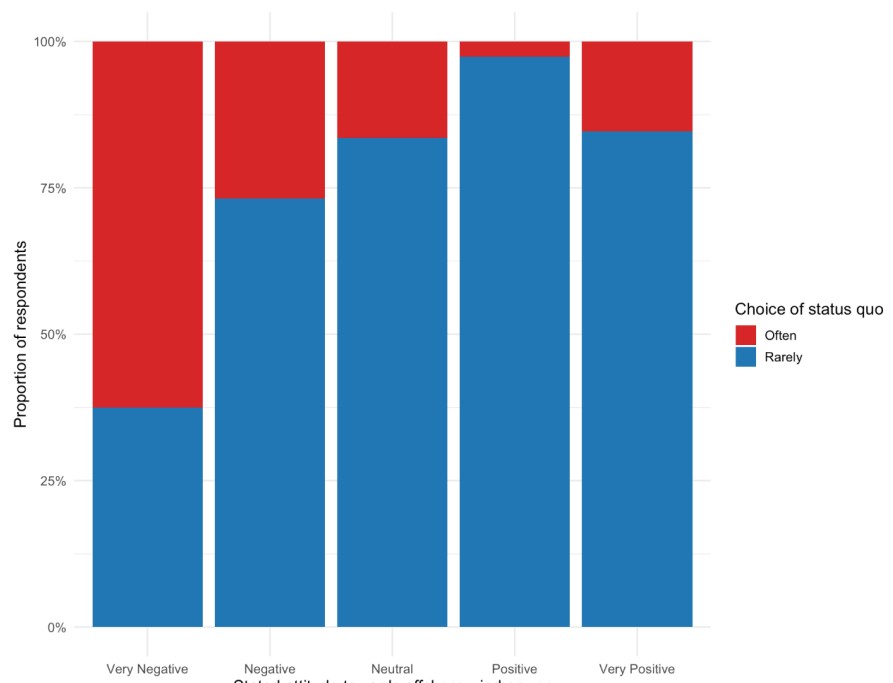

**Figure 6. Relation between attitude toward offshore wind power and *status quo* choice frequency by respondents.**

Table 7 presents the coefficients of the ZINB model and distinguish them between the count component (number of *status quo* choices among respondents capable of selecting it) and the zero-inflation component (probability of always choosing zero). In the count model, the attitude toward offshore wind power was a significant predictor ($\beta$ = 0.21, p < 0.001): it indicates that respondents with a more negative attitude toward offshore wind power were more likely to choose the *status quo*. The corresponding IRR of 1.234 (Table 7) suggests that for each unit increase

in the scale of attitude toward offshore wind power (from very positive to very negative), the expected number of *status quo* choices increases by approximately 23%. Other covariates in the count model including declared gender, age, education, professional status, monthly household revenue, prior knowledge/exposure on OWF, NEP mean score, ocean and professional fishing relationship and distance to the coast were not statistically significant at the 0.05 level.

As the overdispersion parameter ($\theta$) was significantly different from zero ($\log(\theta)$ = 1.282, p < 0.001), it confirms the necessity of a negative binomial specification over another model. The ZINB model revealed that the attitude toward offshore wind power was also a significant predictor of the structural zeros ($\beta$ = -0.508, p < 0.001). This negative coefficient indicates that respondents with a more positive attitude toward offshore wind power are more likely to belong to the group of individuals who never choose the *status quo* over the two other options where the

eco-engineering concept was applied. In other words, the tendency to avoid each time the status quo, and the frequency of status quo choices when selected, are strongly influenced by respondents' attitudes toward offshore wind power.





**Table 7. ZINB model: count and zero-inflation coefficients (IRR for count part).**

| Component | Predictor | Estimate | Standard Error | Z value | p-value | IRR |
|---|---|---|---|---|---|---|
| Count | *Intercept* | 0.019 | 1.231 | 0.016 | 0.988 | 1.019 |
| Count | **Attitude toward OWP** | **0.210** | **0.063** | **3.345** | **0.001\*\*\*** | **1.234** |
| Count | Gender | -0.211 | 0.172 | -1.229 | 0.219 | 0.810 |
| Count | Age | 0.091 | 0.103 | 0.886 | 0.375 | 1.096 |
| Count | Level of education | 0.043 | 0.073 | 0.591 | 0.554 | 1.044 |
| Count | Professional status | 0.056 | 0.042 | 1.342 | 0.180 | 1.057 |
| Count | Monthly household revenue | 0.008 | 0.048 | 0.156 | 0.876 | 1.008 |
| Count | Have already seen OWF | 0.073 | 0.286 | 0.255 | 0.799 | 1.076 |
| Count | Have already heard about OWF | 0.001 | 0.125 | -0.007 | 0.994 | 1.001 |
| Count | NEP mean score | 0.090 | 0.181 | 0.496 | 0.620 | 1.094 |
| Count | Relation to the Ocean | -0.531 | 0.536 | -0.992 | 0.321 | 0.588 |
| Count | Relation to professional fishing | 0.494 | 0.516 | 0.957 | 0.339 | 1.638 |
| Count | Distance to the coast | 0.006 | 0.005 | 1.087 | 0.277 | 1.006 |
| Count | **Log(θ)** | **1.273** | **0.388** | **3.280** | **0.001\*\*\*** | **3.573** |
| Zero | *Intercept* | 1.788 | 0.366 | 4.884 | 0.000 | *NA* |
| Zero | Attitude toward OWP | -0.508 | 0.122 | -4.179 | 0.000 | *NA* |




**Table 8. Reasons of respondents that were exclusive chooser of Option C.**

| **Reasons** | Bouches-du-Rhône | Hérault | Morbihan | Aude | Pyrénées-Orientales | **Total** |
|---|---|---|---|---|---|---|
| The subject (artificial reef) does not interest me. | 0 | 1 | 1 | 1 | 2 | **5** |
| The subject (floating wind turbine) does not interest me. | 2 | 3 | 1 | 0 | 0 | **6** |
| My income is too low. | 5 | 0 | 2 | 0 | 1 | **8** |
| We already pay enough taxes in France. | 13 | 4 | 6 | 0 | 2 | **25** |
| This research is unfeasible. | 4 | 1 | 0 | 1 | 0 | **6** |
| Another reason | 2 | 1 | 2 | 1 | 0 | **6** |
| *Sample (n = )* | *15* | *6* | *8* | *2* | *3* | *34* |
| **Total** | **26** | **10** | **12** | **3** | **5** | **56** |

People that were exclusive chooser of the *status quo* selected mainly the argument of an already too high level of
taxes in France (Table 8) to support the integration of eco-engineering through the electricity bill. Sometimes
people paired their first answer with another reason (13/34 for 2 arguments, 3/34 for 3 arguments), and up to 4
reasons selected (1/34).



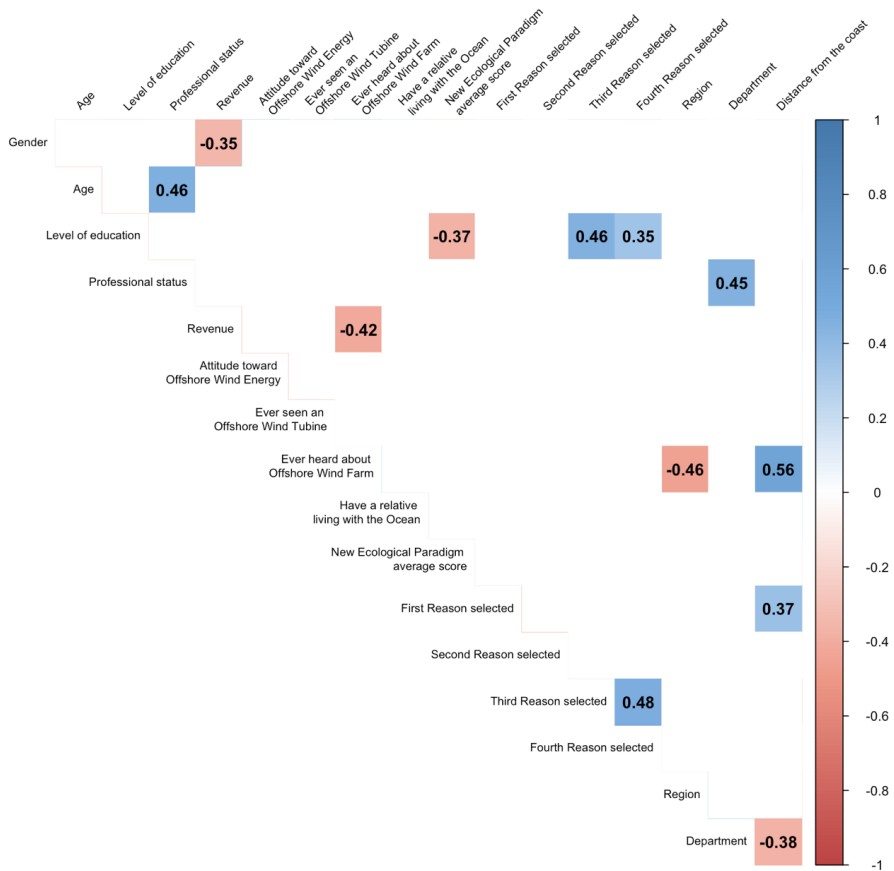

**Figure 7. Correlation plot between parameters of respondents that have only chosen Option C.**

As in section 3.1, a correlation test was performed to analyze the influence of descriptive parameters on the reason

for the frequent choice of the *status quo* (Figure 7). Several notable and significant positive correlations were found such as a correlation (0.37; p-value < 0.05) between the first selected reason of why they always have chosen the *status quo* with distance to the coast from their principal residency. Other interesting negative correlations were found among these respondents, especially between the level of education and their average score on the New Ecological Paradigm (-0.37; p-value < 0.05).





# 470 4 Discussion.

## 4.1 Do preferences vary depending on territories (*i.e.* departments)?

The results indicate a relative homogeneity in individual preferences on the different attributes across the departments sampled, but with one exception: the attribute related to the use of recycled steel. This factor was the only one showing statistically significant variations between territories, which could suggest a territorial

sensitivity to circular economy concerns. Willingness to pay (WTP) estimates (Train, 2009) seems to confirm this pattern with respondents from Bouches-du-Rhône that reported an average WTP of €2.51 for the use of recycled steel (Figure 4 & Table 5; $p < 0.05$) compared to only €0.68 in Hérault. The variation observed for recycled steel may reflect several contextual factors such as differing levels of environmental awareness or education (Joalland and Mahieu, 2023), varying proximity to industrial sectors (*e.g.* steel mills, shipyards) or regional political

narratives emphasizing ecological transition and industrial circularity. However, these interpretations must be qualified by the study's methodological limitations, particularly the small sample sizes for certain departments (*i.e.* Aude n = 20; Pyrénées-Orientales n = 31). The analysis remains less sensitive to subtle variations in such contexts while standard errors were used to account for statistical uncertainty.

In contrast to Lennon et al. (2019) and Perlaviciute et al. (2018), who emphasize contextual diversity and value

pluralism in public acceptability of energy projects, our findings show a marked convergence of citizen preferences. This homogeneity points to a stable core of citizen judgments on key project components, challenging the dominant view of fragmented and context-dependent acceptability. This common ground is particularly strong for attributes such as marine biodiversity and fisheries revenue which showed no meaningful geographic variation across the sample (Figures 3 & 4). This relative consensus in preferences may also stem from the experimental

design itself. Indeed, the choice cards were constructed using attributes identified in a prior study as particularly relevant for coastal populations from different countries (Börger et al., 2015; Dubois et al., 2025a; Klain et al., 2020). Taken together, these findings suggest the presence of a nationally shared vision of support for floating wind technologies that integrate eco-engineering elements. This underlying consensus represents a strategic opportunity for large-scale deployment, especially when combined with a socially acceptable cost-sharing

mechanism such as an electricity bill surcharge. It also leaves room for regionally tailored approaches for accommodating specific cultural, political or informational contexts (Batel, 2020).

Interestingly, these results are consistent with previous international studies showing that preferences for environmental and socio-economic attributes can be remarkably robust across countries, despite significant differences in institutional settings, tax regimes or energy cultures (Firestone and Kempton, 2007; Klain et al.,

2020). This suggests that certain elements, in particular marine biodiversity enhancement and local economic impact, can benefit from broad cross-border support, provided they are properly formulated and culturally significant in the development territory.

## 4.2 Does the attitude toward offshore wind power influence its acceptability?

The Zero-Inflated Negative Binomial model results reveal a statistically significant negative relationship

between the number of *status quo* scenarios selected and respondents' attitudes toward offshore wind energy: individuals with mostly negative views opted for the *status quo* more frequently ($p < 0.001$). This finding contradicts our initial hypothesis, which posited that offshore wind opponents would try to minimize environmental or social impacts by selecting projects that included mitigation measures rather than consistently



choosing the default scenario even though it involved no ecological or socio-economic enhancements. Several
factors may help explain this contradiction. Firstly, it is possible that the *status quo* was perceived as "no project
at all" by respondents, making it a symbolic option for those rejecting offshore wind by principle. Secondly, the
consistency of *status quo* selections (Figure 6 & Table 6) may also indicate a form of systematic refusal, what
some authors refer to as "technology fatigue" or ideology-driven rejection (Anon, 2013; Cohen et al., 2014;
Devine-Wright, 2009). Lastly, follow-up questions revealed that many of these respondents mentioned financial
concerns, particularly regarding the already existing French taxation. Some explicitly stated that they "already
pay too much" and could not support additional fees, even modest ones for eco-engineering regarding an average
electricity bill, suggesting that financial resistance may be tightly bound to broader political or economic
dissatisfaction.

This latter point brings to light a methodological limitation. Although the survey specified that the funding
mechanism was an electricity bill surcharge, many respondents seemed to perceive it differently. This may have
been amplified by a note in the introduction, before the choice sets, stating that the results might be communicated
to policymakers, thus prompting some to use the survey as a space to express discontent with national political
decisions. Despite this, responses from those who consistently selected the *status quo* included thoughtful critiques
such as skepticism about the feasibility of this concept and research offering insights into the multi-layered nature
of rejection, echoing a previous study (Dubois et al., 2025a).

From a practical standpoint, these results emphasize that communication around floating wind projects must go
beyond presenting ecological benefits or compensation measures. It must also engage with deeper social
representations and take into account possible distrust in institutions, perceived unfairness in cost distribution or
even a sense of alienation from decision-making processes (Batel, 2020). These findings are not unique to France.
Previous studies conducted in the United States found similar trends, with respondents rejecting offshore wind for
reasons tied to technological or organization skepticism but also tax fatigue (Dubois et al., 2025a; Firestone and
Kempton, 2007). Likewise, preferences for recycled steel, seen as a symbolic environmental gesture, emerged in
both French and American contexts suggesting that even in culturally distinct settings, some material signals of
"green integrity" carry shared meaning.

These insights are reinforced by other evidence. For example, Iwata et al. (2023) found that Japanese respondents
expressed a negative WTP for offshore wind scenarios that impacted marine species, highlighting a desire for a
global ecological coherence in the context of renewable energy exploitation. The present study aligns with this
pattern as respondents favored scenarios that minimized ecological disruption or offered tangible co-benefits.
Together, these findings suggest a shared normative expectation across countries: renewable energy
infrastructures must not only reduce emissions but also embody a more holistic environmental ethic, in particular
by the primary aim of these technologies, which is to offer more responsible energy. This reinforces the relevance
of nature-inclusive designs (NiDs) that are both technically robust and symbolically credible (Pardo et al., 2023).

Finally, our results echo also prior findings (Klain et al., 2020) showing that choices often reinforce
existing attitudes, instead of changing them. Respondents already skeptical of offshore wind were more likely to
avoid paying extra costs, even for improvements that could deliver broader socio or ecological benefits. This
indicates that values and perceptions may dominate over instrumental logic when acceptance thresholds have
already been crossed. But it's not all doom and gloom, since respondents declaring an "only" negative attitude
(and not a very negative one) nevertheless chose scenarios presenting the option with eco-engineering. This paves



the way for the development of this technology, despite the resistance to wind power sometimes encountered in

coastal communities.

**5 Practical recommendations for policy makers, non-governmental organizations, developers and industry stakeholders.**

The findings of this study offer concrete recommendations for policy makers, non-governmental organizations, developers, consultants and industrial actors engaged in offshore renewable energy deployment.

Public opposition is often rooted not only in technical misunderstandings, but also in symbolic, cultural or emotional dimensions, including distrust toward institutions and perceived loss of democratic agency.

Rather than viewing opposition as a fixed obstacle or a segment to bypass, project developers are encouraged to invest in long-term transparency, credibility and inclusive dialogue. While a portion of the population may appear strongly opposed, our results suggest that "simply negative" attitudes are not always absolute. Many skeptical

individuals remain open to influence, especially when information is relayed by independent and scientifically credible voices, and when projects demonstrate genuine attention to local values (both economic and social) and environmental integrity.

Acceptability must therefore be treated as a core design criterion from the earliest stages of development. This means identifying territories where public dialogue can be constructive, but also developing formats and

partnerships capable of reaching hesitant or mistrustful publics. Collaborative design of eco-engineering features with local stakeholders, as well as regionally tailored, non-promotional communication, can foster trust and shared ownership.

Tools such as Discrete Choice Experiments (DCEs), when complemented with qualitative and deliberative methods, help anticipate public responses and reveal what truly matters to people. Acceptability levers such as

biodiversity enhancement, use of recycled materials or locally economic repercussions should be seen not as "soft" add-ons but as structural components of the project's legitimacy and viability. In a context where environmental legitimacy is no longer presumed, but must be earned, aligning renewable infrastructure with social expectations and durable exploitation is not optional: it is a strategic imperative.

**6 Conclusion.**

The aim of this study was to assess how social preferences for floating wind projects associated with eco-engineering might vary according to territories and stated attitudes towards offshore wind. The survey was designed to capture opinions, visions and voices of non-specialists toward an emerging technology in the world in a state that is not familiar with offshore wind including bottom fixed. The results highlight a relative consistency of preferences across the French coastline subject to the development of this technology. The results also show a

high degree of incidence between negative attitudes towards offshore wind power and the choices made in the experimental scenarios, regardless of ecological or socio-economic metrics.

A close analysis of these results revealed a gradient in behavior. Respondents with "Very negative" attitudes toward offshore wind almost systematically rejected all proposed alternatives, thus preferring the status quo. In contrast, those with simply "Negative" views were more likely to engage with scenarios of applied eco-

engineering. This nuance is essential as it highlights that while a segment of the population may be unreachable through technical or communicative adjustments, another portion remains open to influence when projects are designed with attention to their values and concerns.



Contrary to expectations, compensation and mitigation measures do not always improve project acceptability among opponents. Ideological filters or symbolic interpretations were shown to shape decisions more than the actual attributes presented, especially for people with strong negative attitudes. This underlines the importance of integrating psycho-social dimensions into the study of technological acceptability, particularly during public debate or in the initial phases of a project.

Some limitations of the study should be noted. Uneven sample sizes across departments may have reduced the power of certain local comparisons. Moreover, the hypothetical nature of the scenarios implies a degree of abstraction that may differ from real-world behavior in a concrete policy context. Attitudes were self-reported and may also reflect some social desirability bias. Moreover, no question was asked to monitor participants' attention, to check that they were paying attention to the wording of the question and thus avoid unreflective or "automatic" answers. Future research could explore how emotional factors, risk perception or place-based identity influence preferences with more complex tests such as a Mixed Logit Model.

**Acknowledgments.**

This work is part of the US–French collaborative project Improving the Environmental Integration of Floating Offshore Wind Turbines (I2FLOW) within the Sea and Littoral Research Institute (FR CNRS IUML), Nantes Université, Nantes, France, in partnership with the Ocean Resources and Renewable Energy (ORE) Lab, Department of Mechanical and Industrial Engineering, University of Massachusetts, Amherst, USA, and the Department of Environmental Studies at Colby College, Waterville, Maine, USA. It was funded by Region de la Loire under the WEAMEC community through the MOORREEF project, and the European Community under the FEDER program. During the preparation of this work the authors partially used AI (Quillbot) in order to improve readability, language and grammar of the work. After using this tool/service, the authors reviewed and edited the content as needed and take full responsibility for the content of the published article.

**Authors contribution.**

AD, PAM, AB and FS conceptualised the study. AD, PAM, AB and JM developed the methodology. AD and PAM performed formal analysis, investigation, visualisation. AD, PAM and FS prepared the original-draft. PAM, AB and FS provided supervision. All authors contributed to the draft review and editing.





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





**Table A1. List of Offshore Wind Farms on French maritime territory and their status.**

| Name | Call of tender number (date of launch) | Year of Attribution | Attachment department (number) | Total capacity power | Status (at July 2025) | | Status updating year | Surface (km²) | Technology |
|---|---|---|---|---|---|---|---|---|---|
| Saint Nazaire | AO1 (2011) | 2012 | Loire-Atlantique (44) | 480 MW | Operational | | 2022 | 78 | |
| Saint Brieuc | AO1 (2011) | 2012 | Côtes-d'Armor (22) | 496 MW | Operational | | 2023 | 75 | |
| Fécamp | AO1 (2011) | 2012 | Seine-Maritime (76) | 500 MW | Operational | | 2023 | 60 | |
| Îles d'Yeu et Noirmoutier | AO2 (2013) | 2014 | Vendée (85) | 488 MW | In construction | | 2025 | 83 | |
| Courseulles-sur-mer | AO1 (2011) | 2012 | Calvados (14) | 450 MW | In construction | | 2025 | 50 | |
| Dieppe-Le Tréport | AO2 (2013) | 2014 | Seine-Maritime (76) | 496 MW | In construction | | 2024 | 83 | |
| Dunkerque | AO3 (2016) | 2019 | Nord (59) | 600 MW | Attributed | | 2019 | 50 | Fixed |
| Fécamp Grand Large | AO10 (2025) | - | Seine-Maritime (76) | 2 x 2 GW | Public debate | | 2025 | 483 | |
| Oléron 1 | AO7 (2022) | - | Charente-Maritime (17) | 1 GW | Concurrence | | 2025 | 180 | |
| Oléron 2 | AO9 (2024) | - | Charente-Maritime (17) | 1 - 1.25 GW | Concurrence | | 2025 | 250 | |
| Centre Manche 1 | AO4 (2021) | 2023 | Manche (50) | 1 GW | Attributed | | 2025 | 183 | |
| Centre Manche 2 | AO8 (2022) | - | Calvados (14) | 1.5 GW | Concurrence | | 2025 | 270 | |



| | | | | | | | | |
|---|---|---|---|---|---|---|---|---|
| | | | | | □ | | | |
| Golfe du Lion Centre | AO10 (2025) | - | Hérault (34) | 2 GW | Public debate | ■□□□ | 2024 | 400 | |
| Golfe de Gascogne Sud | AO10 (2025) | - | Charente-Maritime (17) | 1.2 GW | Public debate | ■□□□ | 2024 | 250 | |
| Bretagne Nord-Ouest | AO10 (2025) | - | Finistère (29) | 2 GW | Public debate | ■□□□ | 2024 | 350 | |
| Golfe de Fos 1 | AO6 (2022) | 2024 | Bouches-du-Rhône (13) | 230 - 280 MW | Attributed | ■■■□ □ | 2024 | 52 | Floating (commercial) |
| Golfe de Fos 2 | AO9 (2024) | - | Bouches-du-Rhône (13) | 450 - 550 MW | Concurrence | ■■□□ □ | 2025 | 103 | |
| Narbonnaise Sud-Hérault 1 | AO6 (2022) | 2024 | Aude (11) | 230 - 280 MW | Attributed | ■■■□ □ | 2024 | 48 | |
| Narbonnaise Sud-Hérault 2 | AO9 (2024) | - | Aude (11) | 450 - 550 MW | Concurrence | ■■□□ | 2025 | 96 | |
| Bretagne Sud 1 | AO5 (2021) | 2024 | Morbihan (56) | 250 MW | Attributed | ■■■□ □ | 2024 | 45 | |
| Bretagne Sud 2 | AO9 (2024) | - | Morbihan (56) | 400 - 550 MW | Concurrence | ■■□□ □ | 2025 | 225 | |
| Provence Grand Large/Port-Saint-Louis-du-Rhône | AO ADEME (2015) | 2016 | Bouches-du-Rhône (13) | 25.2 MW | Operational | ■■■■ ■ | 2024 | 0.78 | Floating (pilot) |
| Gruissan | AO ADEME (2015) | 2016 | Aude (11) | 30 MW | In construction | ■■■■ □ | 2025 | 8.15 | |
| Leucate-Le Barcarès | AO ADEME (2015) | 2016 | Pyrénées-Orientales | 30 MW | In construction | ■■■■ □ | 2025 | 6.17 | |





**Appendix 2. The 15 likert-scale (Strongly disagree, disagree, neutral, agree, strongly agree) statements of**
**the New-Ecological Paradigm questionnaire (Anderson & Dunlap, 2012).**

1. We are approaching the limit of the number of people Earth can support.

2. Humans have the right to modify the natural environment to suit their needs.

3. When humans interfere with nature it often produces disastrous consequences.

4. Human ingenuity will ensure that we do not make the Earth unlivable.

5. Humans are seriously abusing the environment.

6. The Earth has plenty of natural resources if we just learn how to develop them.

7. Plants and animals have as much right as humans to exist.

8. The balance of nature is strong enough to cope with the impacts of modern industrial nations.

9. Despite our special abilities, humans are still subject to the laws of nature.

10. The so-called "ecological crisis" facing humankind has been greatly exaggerated.

11. The Earth is like a spaceship with very limited room and resources.

12. Humans were meant to rule over the rest of nature.

13. The balance of nature is very delicate and easily upset.

14. Humans will eventually learn enough about how nature works to be able to control it.

15. If things continue on their present course, we will soon experience a major ecological catastrophe.