# Peer review of "Preference and Willingness-to-pay analysis for an ecoengineering technology for floating wind turbines"

_Wind Energy Science, 2025_

## Referee Comment (RC1)

**Manuscript: "Preference and Willingness-to-pay analysis for an ecoengineering technology for floating wind turbines"**

**General Comments**

The manuscript investigates public preferences and willingness-to-pay for an eco-engineering solution integrated into floating offshore wind farms in France, using a discrete choice experiment across five coastal departments. The topic is timely, relevant, and clearly within the scope of Wind Energy Science, particularly given the accelerating deployment of floating wind and the rising emphasis on social acceptability, marine ecosystem impacts, and nature-inclusive design. The study makes a meaningful contribution by combining a specific engineering concept with a socioeconomic valuation, and by comparing preference heterogeneity across distinct French coastal regions.

The manuscript is generally well written, well structured, and rich in contextual detail. The presentation of background literature, methodological steps, and results is thorough. The econometric analysis is appropriate and the discussion links findings to broader questions of public acceptance and environmental integration of offshore wind. However, some aspects require clarification or tightening before publication.

**Specific Comments**

- 1) The abstract is informative and complete, but overly detailed and thus somewhat lengthy. I believe that a tighter structure and more quantitative reporting would improve its scientific impact. Please consider shortening the context, add one clear sentence identifying the paper's contribution, summarize the method succinctly, and include one or two key quantitative findings (e.g., mean WTP) to make the results more impactful.
- 2) The description of the socio-demographic survey is thorough. However, the authors should specify whether quotas or weighting were applied to approximate regional populations. Otherwise, sample representativeness is uncertain. "As representative as possible" is not sufficient. Please specify sampling weights or quotas used (e.g., age, gender, income).
- 3) The use of "marine biodiversity" and "local fisheries revenue" as attributes may involve correlated perceptions (both positive ecosystem services).
- 4) The design's experimental realism should be discussed. Were visualizations or images used to communicate the concept? If so, please show them in the Appendix for clarity and reproducibility.
- 5) There is no mention of an attention or dominance test (e.g., a consistency check). Without it, internal validity of responses cannot be assessed.
- 6) Discuss potential self-selection bias: Did environmentally concerned individuals over-respond? This can systematically raise WTP.
- 7) The ZINB model is innovative for analyzing status-quo choices. However, variable scaling and coding should be presented in more detail (e.g., attitude scale, income normalization).

- 8) Table 3 is useful, but it could be placed in the Appendix. The paragraph at the begin of the Section suffices to summarize the descriptive statistics.
- 9) The Wald tests are appropriate but under-explained. Please clarify the null hypothesis (pages 14-15, 15-16).
- 10) Figure 6 and Table 6 are informative but repetitive. Please consider keeping in the main text only the Table.
- 11) Cross-references: some tables/figures are cited before their introduction. Please reorder.
- 12) Section 4.1 correctly notes the relative homogeneity of preferences. However, the claim that this "challenges the dominant view of fragmented acceptability" is overstated given the small sample sizes per region.
- 13) The interpretation of recycled-steel preferences as "territorial sensitivity to circular economy" is plausible but speculative. Consider testing correlations with education or employment in industrial sectors.

---

## Author Comment (AC1)

**Manuscript:** « **Preference and Willingness-to-pay analysis for an eco-engineering technology for floating wind turbines** »

**General answer**

We are deeply grateful for your thorough analysis and review of our work on eco-engineering applied to floating wind turbines. We have carefully considered each of your comments and provide detailed responses below. We are convinced that this article is significantly better thanks to your feedback and perspective.

**Answer to the comments of reviewer 1**

1) *The abstract is informative and complete, but overly detailed and thus somewhat lengthy. I believe that a tighter structure and more quantitative reporting would improve its scientific impact. Please consider shortening the context, add one clear sentence identifying the paper's contribution, summarize the method succinctly, and include one or two key quantitative findings (e.g., mean WTP) to make the results more impactful.*

   *Response:* We acknowledge that the initial abstract was overly elaborate and long. In the new version we will condense the contextual material, clearly highlight the paper's major contribution, present the discrete choice experiment (DCE) methodology briefly and report key numeric findings. For instance, we will highlight the marginal willingness-to-pay (WTP) for biodiversity and fishing revenue gains (roughly €0.11 and €0.15 per percentage-point increase, respectively). These figures can provide a concise quantitative insight into the study's primary outcomes.

2) *The description of the socio-demographic survey is thorough. However, the authors should specify whether quotas or weighting were applied to approximate regional populations. Otherwise, sample representativeness is uncertain. "As representative as possible" is not sufficient. Please specify sampling weights or quotas used (e.g., age, gender, income).*

   *Response:* We will clarify the sampling procedure in the revised manuscript. For instance, the company could not formally ensure a balanced distribution of respondents across core socio-demographic factors (i.e. age and gender) at the departmental level. Nonetheless, the final sample composition is broadly consistent with INSEE regional demographic data, providing reasonable confidence in the reliability of the results.

3) *The use of "marine biodiversity" and "local fisheries revenue" as attributes may involve correlated perceptions (both positive ecosystem services).*

   *Response:* The probable perceived correlation between attributes was one of our concerns before starting the survey. We pre-tested the questionnaire on a small group of people but did not find any perceived correlation between "marine biodiversity" and "local fisheries revenue". In the final survey, we nevertheless

broadened the description of the attribute to eliminate any possible perceived link.

4) *The design's experimental realism should be discussed. Were visualizations or images used to communicate the concept? If so, please show them in the Appendix for clarity and reproducibility.*

*Response:* Indeed, visualizations were shown to respondents at the beginning of the questionnaire to explain the notion of artificial reef. Moreover, the specific studied concept of the study was shown and described to respondents. Then, a detailed yet succinct description of each attribute on which the artificial reef will have an influence, and their levels was shown. We will include some visualizations and the choice cards in the appendices, which we will translate into English for the purpose of the paper.

5) *There is no mention of an attention or dominance test (e.g., a consistency check). Without it, internal validity of responses cannot be assessed.*

*Response:* We thank reviewer 1 for raising this important point regarding internal validity. We acknowledge that no explicit attention or dominance test was included. However, we implemented several ex-ante and ex-post checks to support data quality.

- *Ex-ante*: We kept the choice cards short, clearly structured, realistically designed and understandable scenarios. The survey was administered through easypanel, a professional firm we regularly collaborate with which has substantial experience in running DCEs. The company also maintains respondent quality through strict panel management procedures, including double opt-in registration and individual login requirements that prevent link sharing, ensuring that only verified and engaged panel members participate.
- *Ex-post*: We examined response patterns and found very few respondents who systematically chose only option A, only option B, or alternated mechanically (*e.g.*, ABABAB).

We will report these checks in the revised manuscript and acknowledge the absence of an embedded dominance test as a limitation. We did not include such a test to preserve the realism of the tasks and because the D-efficient design implemented, by construction, avoids dominated alternatives. Nonetheless, we agree on the importance of assessing response quality.

6) *Discuss potential self-selection bias: Did environmentally concerned individuals over-respond? This can systematically raise WTP.*

*Response:* Thank you for your comment. We acknowledge not having comprehensive data regarding the direct level of environmental concern of respondents who took part in the survey. To reduce selection bias, however, we mandated that the commissioned company stay vague about the

questionnaire's topic while sending out invitations (*i.e.*, respondents did not genuinely know what the survey was about when they consented to take part). Additionally, the organization takes a number of steps to ensure that no one else completes the responses (a person who is not very interested in the topic cannot ask a relative who is). Overall, the risk of over-representation of respondents interested in the given subject is limited.

7) *The ZINB model is innovative for analyzing status-quo choices. However, variable scaling and coding should be presented in more detail (e.g., attitude scale, income normalization).*

*Response:* We acknowledge the coding of the explanatory variables in the ZINB model was lacking. Table 1 below provides the coding of the explanatory variables. It is slightly different from the one employed in the previous version, so as facilitate understanding and interpretation. The main results are not changed. Only the variable Attitude_toward_offshore remains statistically significant at the 5% level.

Table 1. Explanatory variables in the ZINB model

| Variable name | Type of variable | Codification |
|---|---|---|
| Attitude_toward_offshore | Continuous | From 1 ("very positive") to 5 ("very negative") |
| Age | Continuous | Age of the respondent |
| Gender | Binary | 1 if female, 0 if male |
| Education_level | Binary | 1 if university degree, 0 if no university degree |
| Professional status | Binary | 1 if employed or a student, 0 if unemployed, retired, or other |
| Monthly income (in thousands of euros) | Continuous | Midpoint of the income bracket |
| NEP score | Continuous | Mean NEP score |
| Distance to coast | Continuous | Crow-fly distance (in km) from home to the coast |

8) *Table 3 is useful, but it could be placed in the Appendix. The paragraph at the begin of the Section suffices to summarize the descriptive statistics.*

*Response:* We have taken note of reviewer 1's comment and will place the Table 3 in the appendices to lighten the manuscript.

9) *The Wald tests are appropriate but under-explained. Please clarify the null hypothesis (pages 14-15, 15-16).*

*Response:* We thank the reviewer for this valuable suggestion. The Wald tests were indeed used to assess whether estimated coefficients for a given attribute differed significantly between departments. The null hypothesis ($H_0$) tested in each case was that the coefficients ($\beta$) for the same attribute are equal across departments:

- o $H_0$: $\beta_i(dep_1) = \beta_i(dep_2)$ versus the alternative hypothesis $H_1$: $\beta_i(dep_1) \neq \beta_i(dep_2)$.
  Rejection of the null hypothesis consequently suggests that respondents from two departments valued a given attribute differently. According to our findings, there was just one statistically significant difference between Bouches-du-Rhône and Hérault for the "Recycled steel" attribute ($p < 0.05$). For all other attributes and department pairs, the null hypothesis of equal coefficients could not be rejected, therefore demonstrating rather homogenous preferences across territories. We appreciate the reviewer 1 suggestion and will clarify this null hypothesis explicitly in a revised version to improve transparency.

- o Since the Poe test is frequently used in the literature, we have also utilized it in addition to the Wald-test, to test the null hypothesis of equal distribution across departments, as the Poe test is commonly employed in the literature. The Poe test and the Wald test came to the same conclusions. The p-value of the Poe test or the Wald test will be reported in the amended version manuscript, along with a footnote stating that the findings of the Wald and Poe test are similar.

10) *Figure 6 and Table 6 are informative but repetitive. Please consider keeping in the main text only the Table.*

*Response:* Indeed, both Figure 6 and Table 6 present similar information. In the revised version we will retain only the Table 6 to reduce redundancy, improve readability and enhance access to quantitative data.

11) *Cross-references: some tables/figures are cited before their introduction. Please reorder.*

*Response:* We appreciate reviewer 1 observation. We have checked the cross-references carefully. While most citations appeared to be correctly placed, we note that minor adjustments are necessary to ensure consistency in numbering and ordering in a future revision, as a general reshuffle of the figures based on the comments.

12) *Section 4.1 correctly notes the relative homogeneity of preferences. However, the claim that this "challenges the dominant view of fragmented acceptability" is overstated given the small sample sizes per region.*

*Response:* Indeed, the initial claim regarding the challenge to the dominant view of fragmented acceptability may indeed be over-interpreted given the sample sizes per region. A more moderate interpretation will be adopted in the revised version.

13) *The interpretation of recycled-steel preferences as "territorial sensitivity to circular economy" is plausible but speculative. Consider testing correlations with education or employment in industrial sectors.*

*Response:* We acknowledge that the proposed interpretation linking recycled-steel preferences to "territorial sensitivity to circular economy" remains strong and speculative. Unfortunately, exact employment sector data was not collected in the survey, preventing a direct test of correlations with industrial or manufacturing employment. An interaction variable between the respondent professional activity and the `recycled-steel` attribute can be included in the conditional Logit model. Likewise, an interaction variable between the respondent professional activity and the `recycled-steel` attribute might be introduced. However, none of the interaction variables was statistically significant. An alternate interpretation can possibly be that the preference for recycled materials may represent a broader environmental coherence in respondent beliefs equating "clean" renewable energy with similarly "eco-friendly" products. Thus, we have chosen to present this result more cautiously in the interpretation of territorial differences of the amended version. Finally, even though some sampled departments are industry-oriented (*e.g.,* Bouches-du-Rhône), the ones specialising in steelmaking are not included in the study. There is therefore less of a direct and/or strong link between occupations in this field and the material feature that constitutes the concept of artificial reefs in our study.

Additional references.

Angioloni, S., Cerroni, S., Jack, C., & Ashfield, A., (2024). Eliciting farmers' preferences towards agriculture education in Northern Ireland. The Journal of Agricultural Education and Extension 30(4), 591–615.

Tait, P., Saunders, C., Dalziel, P., Rutherford, P., Driver, T., & Guenther, M. (2020). Comparing generational preferences for individual components of sustainability schemes in the Californian wine market. Applied Economics Letters, 27(13), 1091–1095.

Wakamatsu, M., & Managi, S. (2022). Does spatially targeted information boost the value of ecolabeling seafood? A choice experiment in Japan. Applied Economics, 54(52), 6008–6021.